# Comparison of 2D Optical Imaging and 3D Microtomography Shape Measurements of a Coastal Bioclastic Calcareous Sand

**DOI:** 10.3390/jimaging8030072

**Published:** 2022-03-14

**Authors:** Ryan D. Beemer, Linzhu Li, Antonio Leonti, Jeremy Shaw, Joana Fonseca, Iren Valova, Magued Iskander, Cynthia H. Pilskaln

**Affiliations:** 1Department of Civil and Environmental Engineering, University of Massachusetts Dartmouth, Dartmouth, MA 02747, USA; 2Department of Civil and Urban Engineering, New York University, New York, NY 10012, USA; ll3256@nyu.edu (L.L.); iskander@nyu.edu (M.I.); 3Department of Computer and Information Science, University of Massachusetts Dartmouth, Dartmouth, MA 02747, USA; aleonti@umassd.edu (A.L.); iren.valova@umassd.edu (I.V.); 4Centre for Microscopy, Characterisation & Analysis, University of Western Australia, Crawley 6009, Australia; jeremy.shaw@uwa.edu.au; 5Department of Civil Engineering, City, University of London, London EC1V 0HB, UK; joana.fonseca.1@city.ac.uk; 6School for Marine Science and Technology, University of Massachusetts Dartmouth, Dartmouth, MA 02747, USA; cpilskaln@umassd.edu

**Keywords:** particle shape, microtomography, dynamic image analysis, 2D particle shape, 3D particle shape, granulometry, calcareous, carbonate

## Abstract

This article compares measurements of particle shape parameters from three-dimensional (3D) X-ray micro-computed tomography (μCT) and two-dimensional (2D) dynamic image analysis (DIA) from the optical microscopy of a coastal bioclastic calcareous sand from Western Australia. This biogenic sand from a high energy environment consists largely of the shells and tests of marine organisms and their clasts. A significant difference was observed between the two imaging techniques for measurements of aspect ratio, convexity, and sphericity. Measured values of aspect ratio, sphericity, and convexity are larger in 2D than in 3D. Correlation analysis indicates that sphericity is correlated with convexity in both 2D and 3D. These results are attributed to inherent limitations of DIA when applied to platy sand grains and to the shape being, in part, dependent on the biology of the grain rather than a purely random clastic process, like typical siliceous sands. The statistical data has also been fitted to Johnson Bounded Distribution for the ease of future use. Overall, this research demonstrates the need for high-quality 3D microscopy when conducting a micromechanical analysis of biogenic calcareous sands.

## 1. Introduction

Advanced imaging techniques are used in the area of micromechanics to measure the shape parameters of soil grains on the micro-scale and correlate them to geotechnical properties on the macro-scale [1,2,3,4,5]. Shape parameters such as aspect ratio, convexity, and sphericity are commonly measured with two-dimensional (2D) dynamic image analysis (DIA) and 3-dimensional (3D) X-ray microtomography (μCT). DIA methods rely on imaging the 2D projection of a grain as it falls through the air [6,7,8,9]. This allows for a large number of particles to be sampled and low computational effort during analysis, but it does not measure the shape of the entire grain. In μCT, a series of radiographs of the sample are taken from multiple angles and then reconstructed into a 3D model. Variation in X-ray absorption between single grains or bulk soil and the background provides image contrast [10,11,12,13,14,15]. Although this method provides an accurate representation of the soil grains, it is limited in the number of grains it can image at once and is computationally more intensive. This is especially the case when watershed techniques are used to isolate individual grains for the μCT of bulk sand samples.

These imaging techniques have been especially useful for the study of problematic offshore calcareous sediments. These soils consist mainly of the calcium carbonate skeletal remains of marine microorganisms such as foraminifera, mollusk, coral, bryozoans and their bioclasts. Unlike typical siliceous sand, the shape of the calcareous sand grain is in part biologically driven. Grains in these sediments can consist wholly of shells or tests (e.g., shells of single-celled organisms) and their bioclasts. The complex shape of the grains in these calcareous sands contributes to their poor geotechnical behavior [16,17,18,19,20], such as pile running [21]. Although a few studies have examined the impact of 2D and 3D imaging techniques on the measurements of sand grains [22,23,24], there have been no direct comparisons of 2D DIA methods to 3D μCT of bulk samples. This paper investigates the impact of 2D DIA and 3D μCT imaging methods on the measured shape parameters, aspect ratio, convexity, and sphericity, of calcareous beach sand. In particular, the ability of the two techniques to measure the shape of biogenic platy and shelly features is examined.

## 2. Sand in Study

The Ledge Point coastal bioclastic sand used in this study was obtained from the coast of Ledge Point, WA, Australia (Figure 1) by a team from the University of Western Australia [25]. The sand has a carbonate content of 91% and consists largely of plate grains hollow foraminifera tests, mollusk shells, bryozoans, and their bioclasts. A selection of µCT scans of sand grains rotated through 180° from Ledge Point is presented in Figure 2. A summary of the site and geotechnical properties is provided in Table 1.

## 3. Microscopy Techniques

### 3.1. Dynamic Image Analysis (DIA)

Two-dimensional DIA for determining granular soil particle size and shape distribution has been shown to be feasible, repeatable, and accurate for many soils [7]. The method employs a high frame rate camera combined with a laser to image millions of individual particles in a short time. In this study a QICPIC (Sympatec, Clausthal-Zellerfeld, Germany) was employed to capture images of Ledge Point sand. The device consists of a vibratory feeding system, a dispersing system, and an imaging sensor.

The device operates as follows: (a) a 100 g specimen is poured into the device through a hopper (VIBRI) and sent to the disperser at a constant rate. (b) The specimen disperses as it falls through a 50 cm long fall shaft (GRADIS) towards the imaging sensor. A small amount of sand particles is transported through the GRADIS simultaneously, which maximizes particle separation. As a result, particle overlap is minimized, thereby eliminating the need for particle segmentation. (c) As particles pass through the image analyzer, particle shapes are captured at a frame rate of 175 frames per second with a 4 Megapixel (2336 × 1728) resolution, and the resulting image resolution is 4 µm/pixel. A more detailed description of the procedure is provided in [7]. For this study, the 100 g sample resulted in a sample size of 1,048,575 particles for the 2D DIA method.

### 3.2. X-ray Microtomography

X-ray micro-computed tomography (µCT) is a non-destructive imaging technique that relies on variations in the attenuation of X-rays as they pass through materials of differing density or mass to generate contrast in the resulting images. Multiple radiographs of the sample are taken from a range of angles, and this information is used to computationally reconstruct a 3D volume.

For the µCT of Ledge Point sediments, a 5 mm diameter plastic tube was filled with calcareous material, mounted onto the instrument stage, and then scanned using a Versa 520 XRM (Zeiss, Pleasanton, CA, USA). Imaging was conducted at 50 kV and 4 W using an LE3 source filter to minimize beam hardening and to improve contrast. Source sample and sample detector distances were set to −13 and 127 mm, respectively, which, in combination with the 0.4× objective lens and 2× camera binning, resulted in a final isotropic voxel resolution of 6.4 µm. Suitable image intensity was achieved using a 4 s exposure, and a total of 2501 X-ray projections were collected through 360° for each tomography.

Radiographs were automatically reconstructed using XRM Reconstructor (Zeiss, Pleasanton, CA, USA) using a default center shift and beam hardening corrections.

### 3.3. Three-Dimensional Watershed Segmentation

A watershed segmentation algorithm was used to segment the reconstructed µCT scans (Figure 3a). The watershed algorithm was developed by Kong and Fonseca [14] and adapted for branch recursive processing by Leonti et al. [26] to improve its computational speed. The preprocessing and segmentation steps follow.

The reconstructed µCT images, in Figure 3a, are first binarized using Otsu’s method [27]. The fully encompassed voids are filled. The scan is then partitioned into two images of the same size as the original, one containing objects smaller than a user-specified diameter, with the other containing everything else. Single-pixel artifacts from partitioning are removed, and the objects are filled.

The final preprocessing step before the watershed segmentation process is to split the image into overlapping sections through the *y*-axis. Subdividing the image allows for quicker, higher quality segmentation. The sections must overlap to avoid the hard boundaries that would result from them being disjointed and adversely affect segmentation.

The connected components of a single section are calculated. The connected components are disconnected regions of pixels that do not touch each other. The components are handled iteratively. The negated Euclidean distance map is calculated, which is the distance of any ‘set’ (1) pixel to the nearest ‘off’ (0) pixel. Next, the bring-up method proposed in [14] is used to dampen all local minima in the component (a mechanism for combining shapes when performing watershed segmentation). Finally, watershed segmentation is performed. If the component is split into two or more smaller regions, the segmentation process is recursively performed on each of those until watershed segmentation is constant. Once this termination case is reached, the original component is replaced by its segmented counterparts.

This process is performed on every section until the entire image is segmented. For the overlapping regions, only the latest segmented version is retained. For this study, the 5 mm diameter sample resulted in a sample size of 2325 for the 3D µCT method.

## 4. Shape Parameters

### 4.1. 2D Shape Parameters

The 2D shape parameters used for analysis of particles from the DIA imaging are summarized below and described in Table 2.

EQPC diameter (*d_EQPC_*) in 2D, *d_EQPC_* is the averaged parameter representing the diameter of a circle having an equivalent area to the particle under consideration:(1)$$d_{EQPC}=\sqrt{\frac{4\times A}{\pi }}$$

Feret diameters are the distance between two parallel tangents to the particle at an arbitrary angle [28]. Feret-max and Feret-min diameters (*d_Fmax_*, *d_Fmin_*) are the longest and shortest diameters from one particle image and are generally employed for describing the maximum and minimum dimensions of a particle. 

Three types of particle shape descriptors were employed to quantify particle shape morphology (Table 2). The selected shape descriptors are thought to capture independent shape features [23]. The numerical value of shape descriptors ranges from 0.0 to 1.0, where a symmetrical particle, such as a sphere, approaches 1.0, while a highly irregular particle has descriptors approaching, but never reaching, 0.

Aspect ratio (*AR_2D_*) in 2D is defined as the ratio of the minimum and maximum Feret dimensions [28]:(2)$$AR_{2D}=\frac{d_{Fmin}}{d_{Fmax}}$$

Sphericity (*S_2D_*) in 2D has been defined by ISO 2008 [28] as the ratio of the perimeter of an area-equivalent circle to the real perimeter (*P*),
(3)$$S_{2D}=\frac{\pi \times d_{EQPC}}{P}$$

Convexity (*Cx_2D_*) in 2D is a measure of the overall concavity of a particle [28]. It is the ratio between the particle area (*A*) and the volume of the convex hull (*Ac*) in 2D shape analysis, as shown in Figure 3:(4)$$C_{x2D}=\frac{A}{A_{c}}$$

### 4.2. 3D Shape Parameters

The 3D shape parameter used to analyze the particles from the segmented μCT scans are summarized below and described in Table 2.

ESD diameter (*d_ESD_*) in 3D is the averaged parameter representing the diameter of a sphere having an equivalent volume to the particle under consideration:(5)$$d_{ESD}=\sqrt[{3}]{\frac{6V}{\pi }}$$

Feret-length, Feret-width, and Feret-thickness (*d_Flength_*, *d_Fwidth_*, *d_Fthickness_*) are generally employed for describing the longest, intermediate, and shortest dimensions of a soil particle [29,30]. The traditional definition employs three axes that are always perpendicular to each other. In 3D shape analysis, *d_Flength_*, *d_Fwidth_* and *d_Fthickness_* were obtained with principal coordinate analysis (PCA) through the regionprop3 and pcacov functions in MATLAB ver. R2021b. 

Aspect ratio in 3D is defined from the ratio of the three Feret dimensions. In 3D, there are three different parameters that are typically calculated, including thickness-to-length ratio (*AR_3D_*), elongation index (*EI*), and flatness index (*FI*) [31]. *AR_3D_* is calculated as the ratio between *d_Fthickness_* and *d_Flength_*, and *FI* = *d_Fthickness_*/*d_Fwidth_*, *EI* = *d_Fwidth_*/*d_Flength_*. From this, *AR_3D_* = *EI*∙*FI;* so, only two of these parameters are independent. The definitions of *AR_3D_*, *EI,* and *FI* are adopted for reconstruction of µCT images, while in 2D DIA, aspect ratio is calculated as *AR_2D_* = *d_Fmin_*/*d_Fmax_* for each particle.
(6)$$AR_{3D}=\frac{d_{Fthickness}}{d_{Flength}}$$
(7)$$FI=\frac{d_{Fthickness}}{d_{Fwidth}}$$
(8)$$EI=\frac{d_{Fwidth}}{d_{Flength}}$$

Sphericity (*S_3D_*) in 3D has been defined from [32] as the ratio of the surface area of a volume equivalent sphere to a surface area of a real particle (*A_s_*):(9)$$S_{3D}=\frac{\pi d_{ESD}^{2}}{A_{s}}$$

Convexity (*Cx_3D_*) in 3D is a measure of the overall concavity of a particle and taken from [28], but updated for particle with internal voids by [14]:(10)$$C_{x3D}=\frac{V_{fill}}{Vc}$$

## 5. Results

### 5.1. Particle Size Distribution

Particle size distributions have been compared between 2D DIA, 3D μCT, and mechanical sieve for the bioclastic coastal sand in Figure 4. The sieve was conducted in line with [33], with the sieve sizes listed in Table 3. The particle size distributions from microscopy analyses were created by setting the histogram bin edges to the sieve sizes listed in Table 3. The 2D EQPC diameter aligns well with the mechanical sieve, and the 2D Feret range (*d_Fmin_*–*d_Fmax_*) bounds the sieve. The 3D ESD significantly overestimates the mechanical sieve, the *d_50_* of the sieve being 0.11 mm and the *d_50_* of the ESD being 0.38 mm. The mechanical sieve falls entirely outside the 3D Feret range (*d_Fthickness_*–*d_Flength_*). Although the mechanical sieve cannot be defined directly by a specific analytical description of grain shape, it is clear from Figure 3. that the 3D μCT is underestimating the quantity of grains below 0.2 mm in diameter. It is likely that the segmentation algorithms are unintentionally eliminating smaller grains. During this process, it is difficult to balance over segmentation and the preservation of naturally small grains. Note that the sieve analysis also lacks the precision to capture small particles accurately.

A key limitation of μCT is throughput. While the 3D segmentation of soil grains from μCT could be improved by higher resolution scanning, doing so would reduce the field of view, which would further reduce throughput. Continued development of these watershed techniques, or increased μCT throughput, could improve the 3D segmentation of soil grains from μCT in the future. The particle size distribution may have also been impacted by sample size. The 2D DIA sample size was over 450 times larger by particle count than the 3D μCT.

### 5.2. Shape Parameter Variation with Size

The relationship between particle size and shape parameters for 2D DIA and 3D μCT is compared by calculating the mean particle shape parameters (Figure 5). The particle size distribution is provided as a probability density plot (histogram) as is common in the field of geology. The mean particle shape parameters were calculated from bins where the edges were set at the sieve opening sizes listed in Table 3. This should represent the mean shape parameter being calculated for the material that would be accumulated on each sieve. The 2D DIA shape parameters *AR_2D_* and *Cx_2D_* are larger than the 3D μCT parameters across all sizes. For example, at 0.3 mm: *AR_2D_* = 0.66, *AR_3D_* = 0.44, *Cx_2D_* = 0.88, and *Cx_3D_* = 0.65. All 3D μCT parameters vary with size, with *S_3D_* changing the most. For example, at 0.3 mm: *S_3D_* = 0.63, and at 2.36 mm: *S_3D_* = 0.36. *AR_2D_* varies the most with size, and *S_2D_* shows *Cx_2D_* small variation with size. Below 0.2 mm, the 3D parameters could have been affected by the segmentation algorithm over-filtering the small grains.

### 5.3. Statistics of Particle Shape Parameter

Probability density plots, presented as histograms, of the 2D DIA and 3D μCT shape parameters calculated from Feret dimensions (aspect ratio, elongation index, and flatness index) are presented in Figure 6. The mode of *AR_2D_* is nearly double that of *AR_3D_* and the two histograms skew in the opposite direction. It appears that the project *AR_2D_* value overestimates aspect ratio relative to *AR_3D_* for this coastal calcareous sand. Flatness compares well with 2D DIA aspect ratio, indicating that the projected section of the grains in DIA is better represented by the length and width of the 3D particles, which is reasonable for a platy or shelly particle. Finally, elongation index does not trend well with *AR_2D_*.

Probability density plots or histograms for 2D DIA and 3D μCT convexity and sphericity are presented in Figure 7. The modes of both *Cx_2D_* and *S_2D_* are larger than those of *Cx_3D_* and *S_3D_*, at 35% and 23% percent larger, respectively. The shape of the distributions varies between the two imaging techniques. The *Cx_2D_* and *S_2D_* histograms skew to the left (mean smaller than the median), while in 3D μCT the *Cx_3D_* and *S_3D_* histograms appear to be normally distributed. It appears that convexity and sphericity calculated from the projected shape of a calcareous sand grain is not an accurate representation of the real 3D shape. The measured values in 2D are significantly larger, dependent on grain size (Figure 5), and the shape of distributions are dissimilar.

The histograms of shape parameter for this coastal calcareous sediment are non-normal. This was also seen in [22,23] for the 2D DIA of Ledge Point and the Browse #1 hemipelagic calcareous sand. For this study, the Bounded Johnson Distributions [34] were fitted to all of the shape parameter histograms; fit lines are shown in Figure 6 and Figure 7. The four parameter Johnson family of distribution is identified by [35] as being useful for modeling non-normal geotechnical data. The Bounded Johnson Distribution limits the range of the random variable to 0 ≤ x ≤. 1, which matches the normalized definitions of the particle shape parameters. Its probability density function can be defined by
(11)$$f\left(y\right)=\frac{\delta }{\sqrt{1+{\left(\frac{x-\xi }{\lambda }\right)}^{2}}}\Phi \left(\gamma +\delta \times ln\left(\frac{x-\xi \;}{\lambda +\xi -y\;}\right)\right)$$
where Φ is the normal distribution probability density function, *x* is an independent random variable, *δ* and *γ* are fitting parameters, *ξ* is the location variable, and *λ* is the scaling variable.

The SciPy distribution fitting function [36] was used to obtain the four Johnson fitting parameters for the shape parameters *AR_2D_*, *AR_3D_*, *EI*, *FI*, *Cx_2D_*, *Cx_3D_*, *S_2D_*, and *S_3D_* for the Ledge Point bioclastic calcareous sand in Table 4.

### 5.4. Correlation of Particle Shape Parameters

A correlation coefficient analysis was conducted on both the 2D DIA and 3D μCT shape parameters and average diameters, shown in Table 5 and Table 6, respectively. These results are also presented graphically as correlation plots of sphericity versus aspect ratio and convexity and convexity to aspect ratio for 1000 randomly selected particles (in Figure 8), while correlation plots comparing *Cx_3D_* and *S_3D_* versus elongation index and flatness index for 1000 randomly selected particles are presented in Figure 9. The strongest correlations of shape parameters are of *S_2D_* with *Cx_2D_*, 0.77, *S_3D_* with *Cx*, 0.85, and *FI* with *AR_3D_*, 0.72. These are associated with the tightest grouping of points in Figure 8 and Figure 9. It should be noted that *AR_3D_* = *EI*∙*FI;* so, only two of these parameters are independent, and the correlation of these parameters is reasonable.

## 6. Discussion

The sphericity, aspect ratio, and convexity measured with 2D DIA were significantly larger than that measured with 3D μCT; in particular, *S_2D_* was approximately 66% larger than *S_3D_* at a particle size of 2.36 mm (Figure 5). This difference appears to be a result of measuring shape parameters from the projection of platy particles such as those common in the Ledge Point coastal bioclastic calcareous sediment, shown in Figure 2.

A thought experiment can be used to examine the possible sets of shape parameters such as *AR_2D_* and *S_2D_* from platy calcareous sand grains. Consider a thin disc with a radius of one unit that rotates at an angle *θ* about the *y*-axis, as in Figure 10. As the disc rotates, its projected shape will go from a disc to a line. The 2D projected aspect ratio of the thin disc at any angle *θ* will be
(12)$$AR_{disc}=\cos\left(\theta \right)$$
where *AR_disc_* is the aspect ratio of the projected disc, and *θ* is the rotation of the disc about the *y*-axis (Figure 10).

The equivalent diameter of a circle with the same area as the project disc will be defined by
(13)$$d_{EQPC-Disc\;}=\sqrt{4\times \cos\left(\theta \right)}$$
where *d_EQPC-Disc_* is the equivalent diameter of a circle with the same area as the projected disc.

The perimeter of the projected disc with a radius of one unit can be estimated using [37] the perimeter of an ellipse formula: (14)$$P=\pi \left[3\left(a+b\right)-\sqrt{\left(3a+b\right)\left(a+3b\right)}\right]$$

The sphericity of the projected disc can then be calculated from Equations (3), (12) and (13) with the major radius of the projected disc being *a* = 1.0 and the minor radius of the projected disc being *b* = cos (*θ*).
(15)$$S_{disc}=\frac{\sqrt{4\times \cos\left(\theta \right)}}{3\left(1+\cos\left(\theta \right)\right)-\sqrt{\left(3+\cos\left(\theta \right)\right)\left(1+3\times \cos\left(\theta \right)\right)}}$$
where *S_disc_* is the sphericity of the projected disc.

The set of possible *AR_disc_* and *S_disc_* for a disc with a radius of one unit rotated at an angle θ about the *y*-axis are plotted in Figure 11. It can be seen that both *S_disc_* and *AR_disc_* follow a sinusoidal pattern. Sphericity has very wide peaks, with the value being close to one for nearly half of the set of angles.

The 2D DIA imaging of a grain falling in front of an imaging sensor at a random angle can be modelled by randomly sampling Equations (12) and (15) at 10,000 random angles θ about the *y*-axis and plotting the probability density, as in Figure 12. Rotation about the x-axis would follow the same formulations presented above due to the disc symmetry. Rotations about the *z*-axis will have no impact of the projected shape. The 2D DIA method process of measuring the shape parameters from the projection of grains clearly overestimates both aspect ratio and sphericity of platy particles, with sphericity being the most extreme, as in Figure 12. The mode of both parameters is approximately one, while both *AR_3D_* and *S_3D_* would approach zero for very thin discs (for a diameter to thickness ratio of 10:1 *S_3D_* = 0.46, which is similar to that of the larger diameter grains, shown in Figure 5c). This aligns well with the results of the 2D DIA and 3D μCT of the Ledge Point coastal bioclast calcareous sand (Figure 5, Figure 6 and Figure 7). In addition, this example demonstrates that any error in the 2D DIA is inherent to the method itself and will not be corrected through increasing sample size.

The measured convexity from 2D DIA was also significantly larger than that from 3D μCT for particle sizes above 0.2 mm. The impact of using a 2D projection to measure convexity is not as simple to mathematically model as aspect ratio or sphericity; however, a qualitative visual assessment of the mollusk shell, Particle D, in Figure 2 demonstrates how the 2D DIA method can overestimate convexity. Figure 2 presents the 3D μCT scans of a number of Ledge Point calcareous sand grains rotating about their vertical axis (relative to the figure orientation). It can be seen that the projected shape of the mollusk shell has a high convexity as it is rotated, but it is obvious that a shell of this nature is concave in 3D (a low value of convexity). A similar pattern holds for Particle A, which is a shell bioclast. It is therefore likely that the high measured value of convexity in the 2D DIA are the result of the microscopy technique itself.

Three-dimensional sphericity is likely correlated to convexity due to the biomorphology of the sand grains. Intact shells such as that in Figure 2d will have a low sphericity and convexity. As they break down through clastic processes their Feret dimensions should tend towards unity (likely due to the shell thickness) and as a result their sphericity and convexity will increase. Since new grains are constantly being introduced to the sand through biogenesis there will be large young shells with low sphericity and convexity and small older bioclasts with high sphericity and convexity in the system. This aligns well with the variations in *Cx_3D_* and *S_3D_* seen in Figure 5b,c.

The correlation of sphericity to convexity in 2D could be the result of 2D projection angle. For example, when considering the bioclast in Figure 2a angular orientation with the highest sphericity appears to have the highest convexity and vice versa. This would explain how *Cx_2D_* and *S_2D_* can be correlated (Figure 8c), but *Cx_2D_* does not show the same variation in size that *S_2D_* does (Figure 5b,c). The Correlation of sphericity with convexity in 2D DIA methods have also been shown by [22] and [24] for both siliceous and calcareous sands.

Li et al. [22] argued that 3D μCT should be used to assess the shape parameters of calcareous sediments in order to resolve internal voids within the grains and that DIA methods can result in dimensionality projection errors that make particles appear larger in projection (this type of error would have a minimum impact on thin platy particles). The angular projection error presented here for biogenic platy and shelly particles that results in a statistical increase in the measured aspect ratio and sphericity in DIA provides an additional reason for using μCT for measuring the shape parameters of complex calcareous sands.

## 7. Practical Significance of This Study

Calcareous sediments are known for being problematic soils [20,21]. One significant issue is they are known for being unclassifiable. Grain size distribution and calcium carbonate content are not good predictors of geotechnical behavior [38]. Clark and Walker [39] is the most commonly used geological classification scheme for calcareous sediments, but it does not provide direct insights into geotechnical behavior. Thus, there is an industry need for a standard geotechnical classification system for calcareous soils [38,40,41]. Their problematic behavior has been attributed to the unique shape of their grains [20,41]. This study demonstrates the need for researchers to carefully consider the imaging methods used to measure shape parameters and the biomorphology of soil grains when undertaking studies on the mechanical behavior and classification of calcareous sediments.

## 8. Conclusions

This article presents the results of a comparison study of 2D DIA and 3D μCT analyses of a coastal bioclastic calcareous sediment from Ledge Point, Western Australia. This study provides unique insights into state-of-the-art soil imaging techniques for quantifying the particle size and shape of bioclastic calcareous sands. The following conclusions from the study can be drawn:For this calcareous sand, 2D DIA correlates better to the traditional sieve analysis than 3D μCT, as shown in Figure 4. The μCT analysis underestimates the number of fine sand grains below 0.2 mm relative to the sieve test. This is possibly due to the watershed algorithm used for segmenting the sand, which digitally removes smaller grains. Alternatively, it may be due to sampling error arising from the limited imaging volume captured by the μCT device compared with the 2D DIA technique.The 2D DIA mean particle shape parameters aspect ratio, sphericity, and convexity with size were significantly larger (dependent on grain size) than those from 3D μCT (Figure 5).The 3D μCT imaging technique is a more accurate method for measuring particle shape parameters of a bioclastic calcareous sand. When measured in 3D, the grains had a lower aspect ratio, *AR_3D_* vs. *AR_2D_*; had a lower convexity, *Cx_3D_* vs. *Cx_2D_*; and had a lower sphericity, *S_3D_* vs. *S_2D_*, as shown in Figure 6 and Figure 7. This agrees with the visual assessment of the randomly selected grains (Figure 2).A simple analytic/statistical analysis of a disc rotating about a single axis indicates that 2D DIA inherently overestimates the aspect ratio and sphericity of platy particles, as shown in Figure 10, Figure 11 and Figure 12.As demonstrated in Figure 2, 2D DIA is limited in its capabilities to accurately measure the convexity of platy bioclasts and shells. It is possible for a particle, such as that in Figure 2d, to be concave in 3D and its 2D projection not to be.Non-normal Johnson Bounded distributions fit the histograms of 2D and 3D particle shape well (Figure 6 and Figure 7). The fitted Johnson variables have been provided in Table 4.Sphericity (*S_2D_* and *S_3D_*) is correlated with convexity (*Cx_2D_* and *Cx_3D_*), aspect ratio (*AR_2D_* and *AR_3D_*), elongation index (*EI*), and flatness index (*FI*) (Figure 8 and Figure 9). This is likely due to the biogenic nature of the soil in the case of 3D measurements and the imaging method in 2D measurements.

## Figures and Tables

**Figure 1 jimaging-08-00072-f001:**
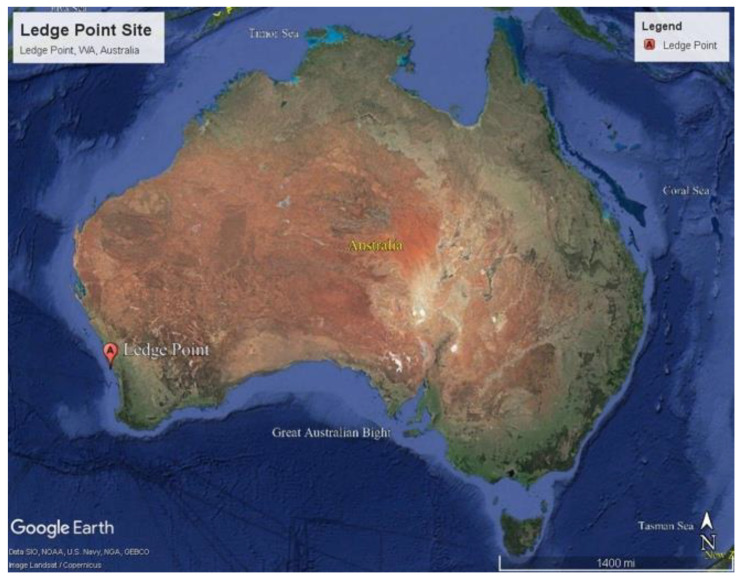
Ledge Point calcareous sand site location.

**Figure 2 jimaging-08-00072-f002:**
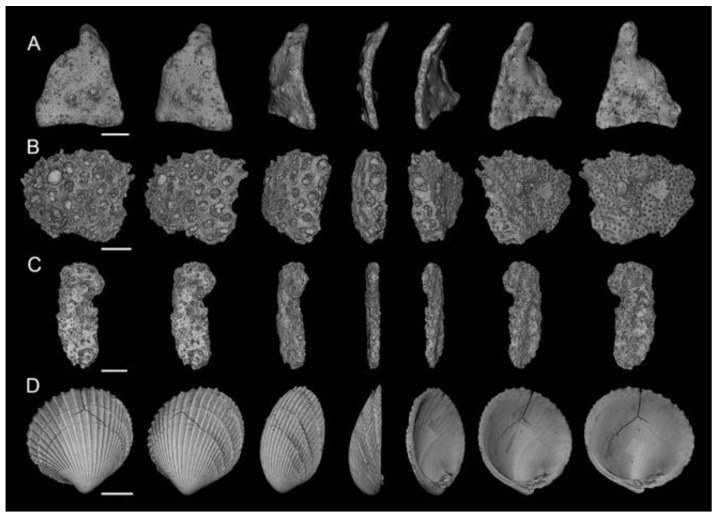
Three-dimensional rendered X-ray µCT images demonstrating typical platy particles from Ledge Point rotated though 180°: (**A**) Bioclast, (**B**) Bryozoan, (**C**) Bioclast, and (**D**) mollusk shell.

**Figure 3 jimaging-08-00072-f003:**
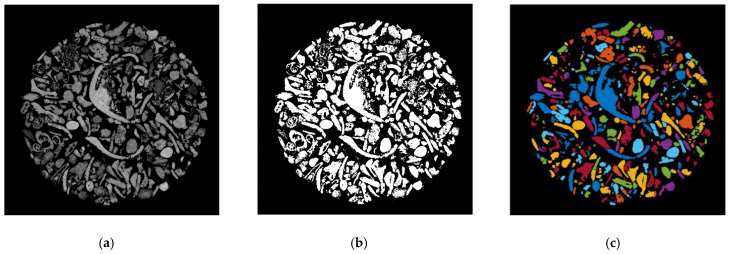
Example of watershed processing of 3D μCT: (**a**) grayscale reconstruction of the µCT slice, (**b**) binary image of the µCT slice, and (**c**) fully segmented image of the µCT slice.

**Figure 4 jimaging-08-00072-f004:**
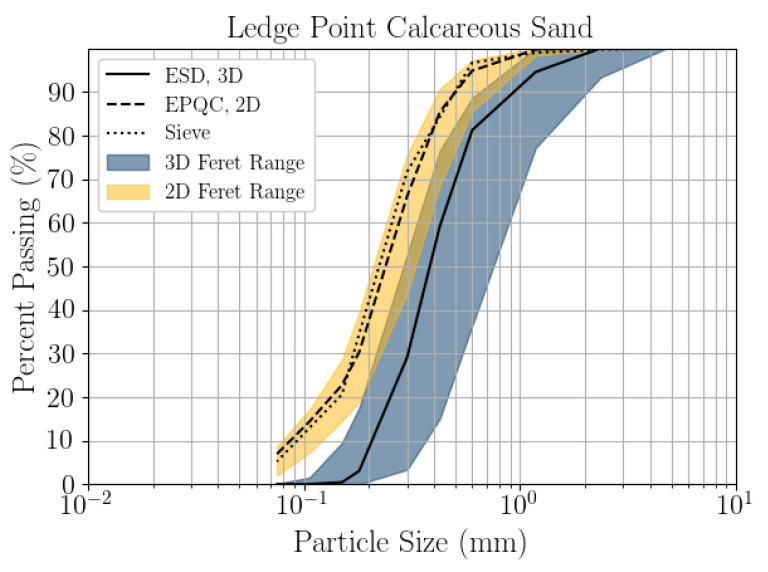
Particle size distribution for Ledge Point from 3D μCT and DIA.

**Figure 5 jimaging-08-00072-f005:**
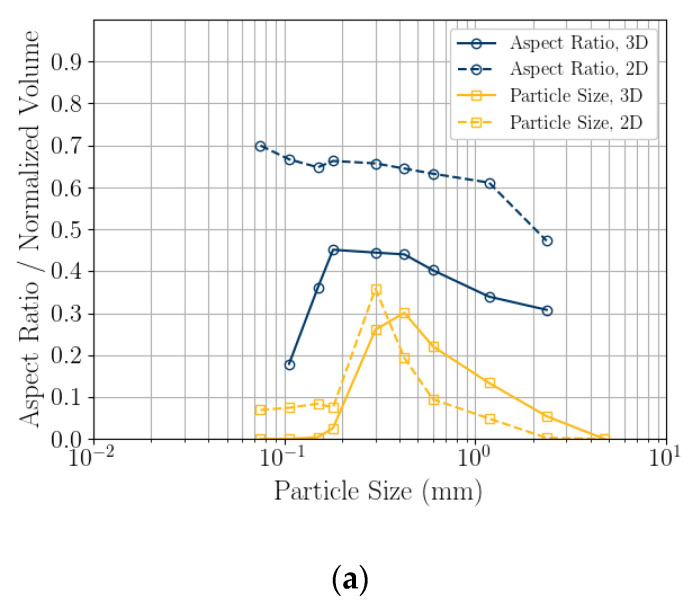
Effect of particle size on the mean particle shape: aspect ratio (**a**), convexity (**b**), and sphericity (**c**). The particle size distribution is provided as a normalized probability density plot, histogram. The mean particle shape parameters were calculated from bins where the edges were set at the sieve opening sizes listed in Table 3.

**Figure 6 jimaging-08-00072-f006:**
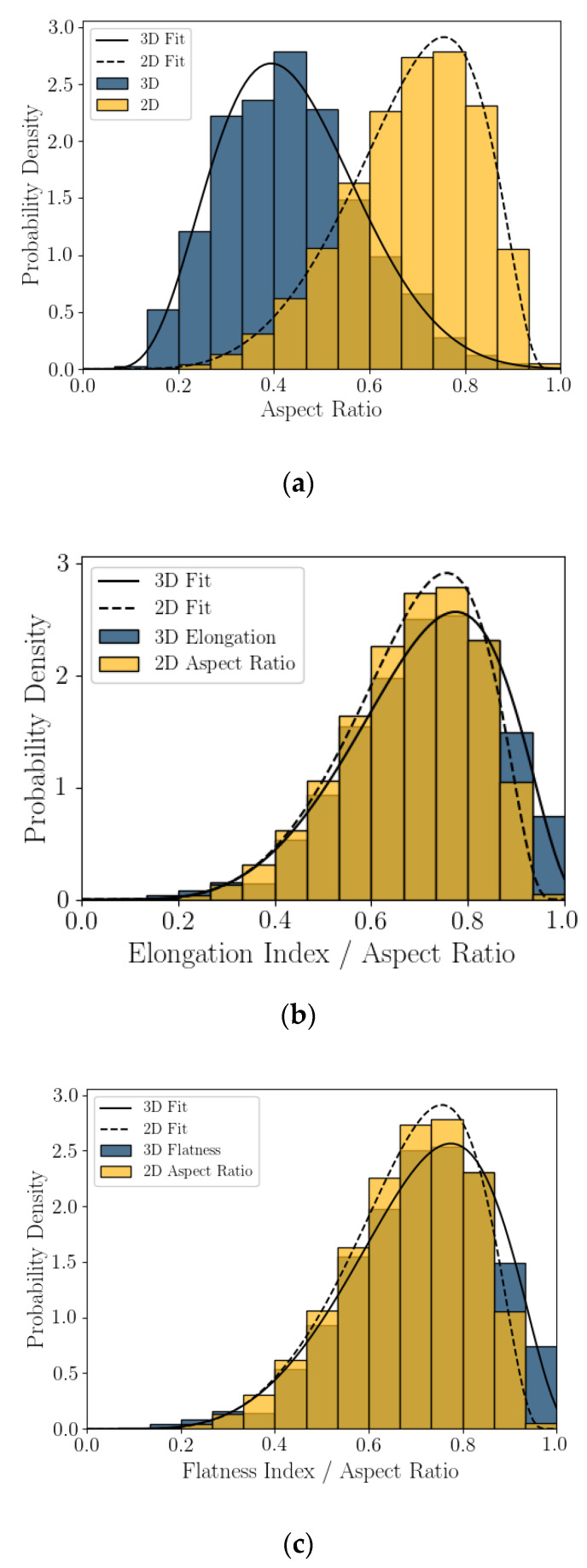
Three shape parameters derived from Feret dimensions: aspect ratio (**a**), elongation index (**b**), and flatness index (**c**), with Johnson Bounded Distribution fitting.

**Figure 7 jimaging-08-00072-f007:**
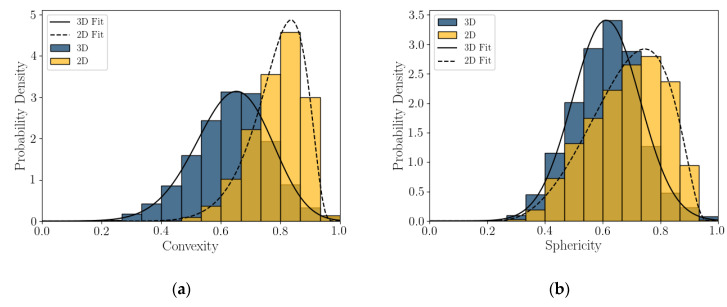
Two-dimensional and 3D convexity (**a**) and sphericity (**b**), with Johnson Bounded Distribution fitting.

**Figure 8 jimaging-08-00072-f008:**
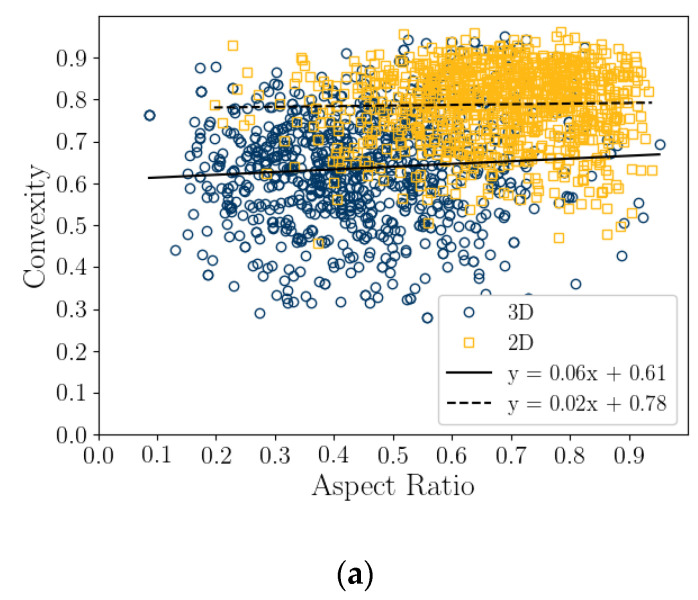
Correlation plots of 2D DIA and 3D μCT aspect ratio, convexity, and sphericity for 1000 randomly selected particles. Convexity versus aspect ratio (**a**), sphericity versus aspect ratio (**b**), and sphericity versus convexity (**c**).

**Figure 9 jimaging-08-00072-f009:**
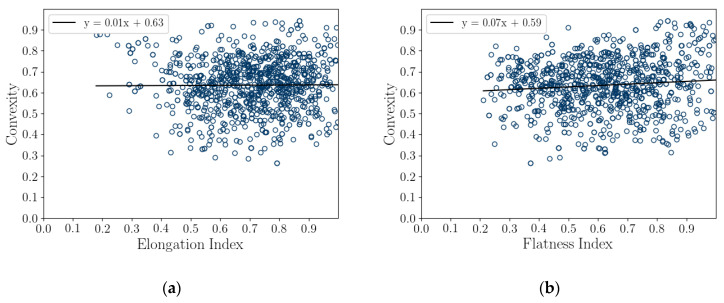
Correlation plots of 3D Feret dimension shape parameter with sphericity and convexity for 1000 random grains. Convexity versus elongation (**a**), convexity versus flatness (**b**), sphericity versus elongation (**c**), and sphericity versus flatness (**d**).

**Figure 10 jimaging-08-00072-f010:**
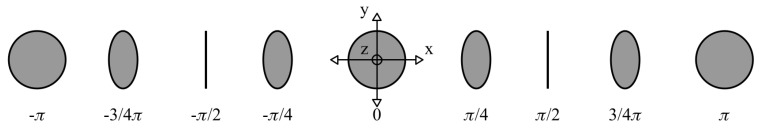
Sketch of rotating disc that demonstrates change in aspect ratio and sphericity.

**Figure 11 jimaging-08-00072-f011:**
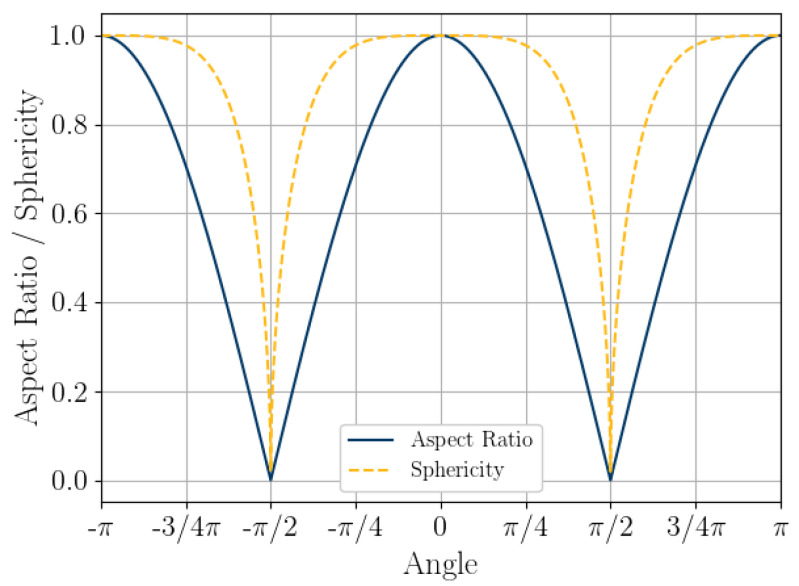
Aspect ratio, *AR_2D_*, and sphericity, *S_2D_*, of a disc rotated about an axis.

**Figure 12 jimaging-08-00072-f012:**
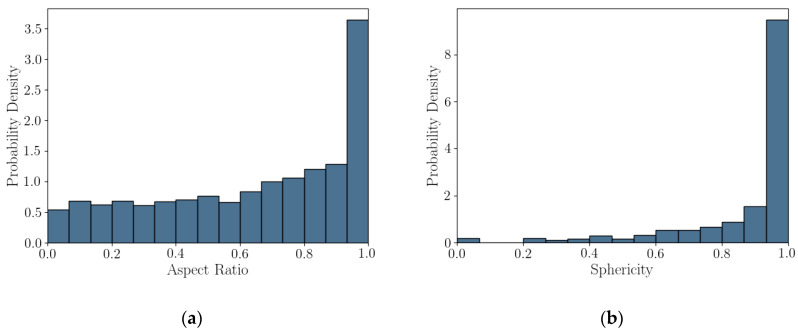
Particle shape parameter histograms of a disc rotated about an axis sampled at 10,000 random angles. (**a**) Aspect ratio and (**b**) sphericity.

**Table 1 jimaging-08-00072-t001:** Ledge point sand site and geotechnical index properties (Sharma 2004).

Name	Location	Water Depth m	*D_50_* µm	CaCO_3_ %	*e_min_*	*e_max_*
Ledge Point	Ledge Point, WA, Australia	0	270	91	0.90	1.21

**Table 2 jimaging-08-00072-t002:** Two-dimensional and 3D shape parameter descriptions.

		DIA	Formula/Explanation			µCT	Formula/Explanation	
Size Descriptors	EQPC (*d_EQPC_*)	Diameter of circle with equivalent particle area	$\sqrt{\frac{4A}{\pi }}$	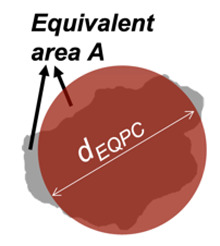	ESD (*d_ESD_*)	Diameter of sphere with equivalent particle volume	$\sqrt[{3}]{\frac{6V}{\pi }}$	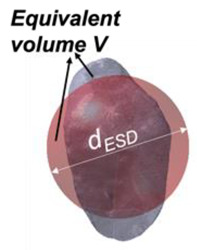
	Feret-max (*d_Fmax_*)	Maximum dimension of a particle, aka. Maximum Feret diameter	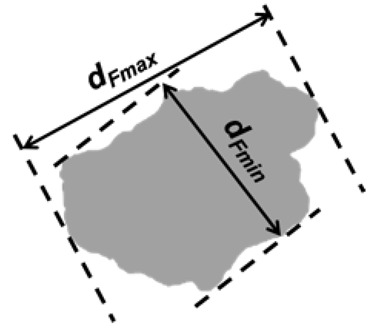		*d_Flength_*	Longest axis	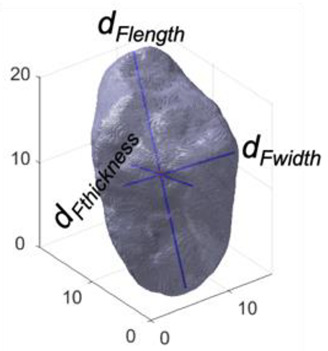	
	Feret-min (*d_Fmin_*)	Minimum dimension of a particle, aka. Minimum Feret diameter			*d_Fwidth_*	Intermediate axis		
					*d_Fthickness_*	Shortest axis		
Shape Descriptors	Aspect ratio (*AR_2D_*)	Ratio of Feret-min to Feret-max	$\frac{d_{Fmin}}{d_{Fmax}}$	—	*AR_3D_*		Ratio of shortest to longest axes	$\frac{d_{Fthickness}}{d_{Flength}}$
					Flatness index (*FI*)		Ratio of shortest to intermediate axes	$\frac{d_{Fthickness}}{d_{Fwidth}}$
					Elongation index (*EI*)		Ratio of intermediate to longest axes	$\frac{d_{Fwidth}}{d_{Flength}}$
	Sphericity (*S_2D_*)	Ratio of the perimeter of a circle with equivalent area to the real particle perimeter	$\frac{\pi d_{EQPC}}{P}$	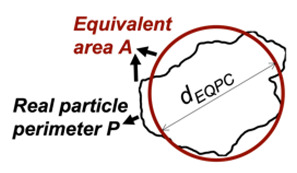	Sphericity (*S_3D_*)	Ratio of the surface area of a volume equivalent sphere to the real particle surface area	$\frac{\pi d_{ESD}^{2}}{A_{s}}$	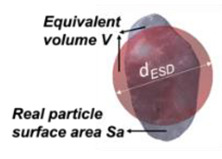
	Convexity (*C_x2D_*)	Ratio between the particle area and the area of its convex hull	$\frac{A}{A_{C}}$	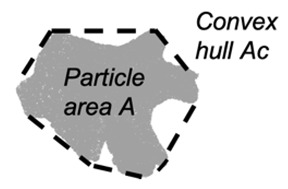	Convexity (*C_x3D_*)	The ratio of the particle volume and the volume of its convex hull	$\frac{V_{fill}}{V_{C}}$	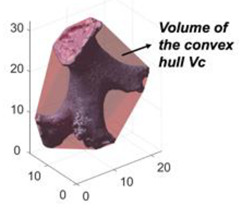

**Table 3 jimaging-08-00072-t003:** Mechanical sieve opening diameter and numerical bin edges.

Sieve Size (mm)
4.75
2.36
1.18
0.60
0.425
0.30
0.180
0.15
0.106
0.075
0.0

**Table 4 jimaging-08-00072-t004:** Johnson Bounded Distribution fitting parameters.

	Sphericity (*S_2D_* or *S_3D_*)	Convexity (*Cx_2D_* or *Cx_3D_*)	Aspect Ratio (*AR_2D_* or *AR_3D_*)	Flatness (*FI*)	Elongation (*EI*)
2D Johnson Bounded Distribution					
*γ:*	−0.8053	−2.2276	−1.1976	—	—
*δ:*	1.2594	1.6607	1.3674	—	—
*ξ:*	0.1809	0.0999	0.0294	—	—
*λ:*	0.7935	0.8872	0.9557	—	—
3D Johnson Bounded Distribution					
*γ:*	−2.102e6	−4.3393	1.3406	−0.0989	−1.0627
*δ:*	1.133e7	4.9303	1.8807	1.0247	1.4269
*ξ:*	−2.92136	−1.5005	0.0001	0.1603	0.0095
*λ:*	5.348e6	3.0300	1.2916	0.8958	1.0573

**Table 5 jimaging-08-00072-t005:** Two-dimensional shape parameter correlation coefficients.

		EQPC	Aspect Ratio	Convexity	Sphericity
		(*d_EQPC_*)	(*AR_2D_*)	(*C_x2D_*)	(*S_2D_*)
EQPC	(*d_EQPC_*)	1	—	—	—
Aspect ratio	(*AR_2D_*)	−0.12	1	—	—
Convexity	(*C_x2D_*)	0.35	0.02	1	—
Sphericity	(*S_2D_*)	−0.06	0.24	0.77	1

**Table 6 jimaging-08-00072-t006:** Three-dimensional shape parameter correlation coefficients.

		ESD	Aspect Ratio	Elongation	Flatness	Convexity	Sphericity
		*d_ESD_*	(*AR_3D_*)	*EI*	*FI*	(*C_x3D_*)	(*S_3D_*)
ESD	(*d_ESD_*)	1	—	—	—	—	—
Aspect ratio	(*AR_3D_*)	−0.07	1	—	—	—	—
Elongation	*EI*	0.03	0.47	1	—	—	—
Flatness	*FI*	−0.10	0.72	−0.25	1	—	—
Convexity	(*C_x3D_*)	−0.23	0.08	0.01	0.09	1	—
Sphericity	(*S_3D_*)	−0.48	0.26	0.02	0.28	0.85	1

## Data Availability

Source data are available from the corresponding author upon reasonable request.

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
