# Peer review of "Comparison of 2D Optical Imaging and 3D Microtomography Shape Measurements of a Coastal Bioclastic Calcareous Sand"

_2313-433X, 2022, doi:10.3390/jimaging8030072_

Round 1

Reviewer 1 Report

I have reviewed this manuscript comparing shape measurements from 2D optical and 3D μCT imaging of a coastal bioclastic calcareous sand. Overall, the manuscript is well written, and the topic is of direct interest to this special issue, as comparing 2D and 3D imaging methods for particle characterisation is significant both from a geotechnical and an imaging standpoint. I raise some points for improvement for the consideration of the authors before this work is considered for publication.

Main points:

o In the abstract, it is claimed that “The shape parameters measured with both methods also show strong correlation with sphericity”, which is not evidenced by Tables 5 and 6. In these tables, 2D sphericity shows a statistical correlation only to convexity (I cannot think of a possible physics/geology-based causality though; can you?), while 3D sphericity only shows a statistical correlation to the Aspect Ratio (which is more convincing, as equant/compact particles tend to take both higher 3D Sphericity and Aspect Ratio values) and a smaller one with the reported Flatness. I find this claim in the abstract ambitious, as all the other indices show nearly no correlation to sphericity.

o I see DIA being used both as “Digital” and “Dynamic” Image Analysis in the manuscript (e.g. sections 1 and 2); is this intentional? I think “Dynamic” is used more frequently for systems like QICPIC, CAMSIZER etc.

o Line 150: The authors cite the explanation of Wadell’s 3D Sphericity, i.e. “ the ratio of the surface area of a volume-equivalent sphere to the surface area of the real particle”, while claiming this is 2D sphericity. 2D projections of particles do not have a volume to begin with. Please correct this, consulting Eq. 9 in Wadell (1935), where a 2D sphericity is proposed, using though the ratio of two diameters, not surface areas (such as in his 3D degree of true sphericity). Please make sure the definition of S_2D you have in your Eq. 3 matches the reference (or feel free to change the reference if needed – many 2D sphericities have been proposed, e.g. see Rorato et al, 2019 or Zheng and Hrysiw, 2015. The latter reference explains the 2D sphericity of Wadell correctly). At the moment, you attribute to Wadell (1935) a definition that he did not propose.

o Line 155: This must be a typo: “and the volume of the convex hull (Ac) in 3D shape analysis, as shown in Fig. 3”. Please make sure you calculate C_x2D as a ratio of areas. Considering this is a 2D parameter used to characterise the results from DIA, I do not see the link between DIA and Fig. 3 showing a slice of the 3D μCT results. Please explain or amend accordingly.

o Line 164: The three Feret dimensions are reported to be perpendicular (belonging to a bounding box) but are illustrated as non-perpendicular in Table 2. To get comparable results, I expect the Feret dimensions to follow the same rules in 2D and 3D shape characterisation. Please provide a reasoning if this is not the case in your analysis.

o Line 165: Please use consistent symbols for your parameters to avoid confusion. At the moment, there are “dFlength, dFwidth, dFthickness” and “dF-length, dF-width and dF-thickness”.

o Line 166: Report which method you used to calculate the minimum bounding box that is used to calculate the Feret dimensions, e.g. did you use the “Rotating Calipers” method of O’Rourke, something similar or PCA? PCA is known to not lead to a minimum box so please clarify which technique was used. I would expect the same methods to be used in the 2D and 3D analysis or else please document the differences for the readers. Also, was a particular software used to calculate the bounding box or did the authors use some in-house code (e.g. the algorithms in Fonseca, 2011)? It would be interesting for the reader if you demonstrated more information on the methods/tools used to derive your results, as shape characterisation results are very subjectively dependent on the methods and assumptions used to calculate them.

o Line 167: Please rephrase; here you use AR_3D to refer to all three aspect ratios in three dimensions, while later on, AR_3D is only used to refer to one of them (AR_3D=thickness/length).

o Line 169: You define AR_TL but not use it. Instead, you use AR_3D. Please keep one of these terms.

o Line 170: There is some confusion on which ratio represents flatness and which elongation. Flatness is equal to the smallest dimension over the intermediate one (Fl=thickness/width), and Elongation is equal to the intermediate dimension over the longest one (El=width/length). Please ensure the validity of your calculations reported in the manuscript and if a mistake has been made, please update your graphs to reflect which parameter is plotted and the discussion. There exists confusion around these indices in the literature (e.g. Clayton, 2009 makes a typo regarding these indices), but e.g. Blott and Pye (2008) make clear what each index represents.

o Line 171: “3D DIA analysis”: This must be a typo, as DIA provided by QICPIC is inherently 2D.

o Equations 6-8: Please revisit your notations and the formulae for flatness/elongation as discussed above. Currently, you report “Ll” instead of “Fl”. Also amend the formulae of El, Fl in Table 3.

o Section 5.2: Comparisons of 2D and 3D sphericity are not straightforward, as different/non-equivalent formulae are used to measure it. E.g. 3D sphericity is calculated here as a ratio of particle volume to the volume of a circumscribed sphere, while 2D sphericity is considering a ratio of perimeters. The latter would be more closely comparable/equivalent to the definition of Wadell for 3D Sphericity, involving the ratio of the surface area of a sphere with equal volume to the particle surface area. Please justify why these parameters are preferred instead of indices that would be intuitively more comparable (e.g. using a sphere/circle with the same area/volume or else using a circumscribed sphere/circle, both in 2D/3D).

o Table 2: The AR_3D index is missing from the table. It is defined in Eq. 6, but so are the rest of the indices shown in Table 2.

o Table 2: In the 3D Sphericity graphic, the equivalent volume sphere is shown, while the definition given in Equation 9 states the circumscribed sphere is used. Please clarify which sphere is considered to calculate 3D sphericity and amend accordingly your calculations, equations and/or graph. Also, the text description of S_3D in table 2 corresponds to the Wadell definition, not matching Equation 9 where a circumscribing sphere is used, as in Kong and Fonseca (2018).

o Lines 221-225: You describe how AR_3D compares to AR_2D, El and Fl, but an explanation of why is lacking. Please provide a deeper discussion. For example, dFmin and dFmax in 2D are not guaranteed to correspond to the real smallest and largest particle axes, and thus the 2D Aspect ratio could be reporting any of the ratios dFthickness/dFwidth, dFwidth/dFlength, dFthickness/dFlength, or other (as both the nominator and denominator are ambiguous), which could by a reason why they correlate somehow with the 3D Flatness and Elongation. I am convinced by your results that 2D characterisation is not adequate even if millions of particles are used; please convince me also with the reasoning of why this is the case.

o Please note that from Aspect Ratio, Flatness and Elongation, only 2 of these indices are independent. In particular, Aspect Ratio = thickness/length = thickness/width * width/length = Fl * El. In case you find this useful, another Aspect Ratio has been used in the literature (e.g. see https://doi.org/10.1007/s11440-021-01362-y), as the average of Flatness and Elongation, i.e. (Fl+El)/2, which in your case might bridge the gap between 2D and 3D measurements. Though, the later is not required by the reviewer, as this paper makes clear the need for 3D measurements when dealing with such complex grains.

o The Discussion section is clear and the thought experiment is excellent in explaining simply how DIA fails to capture the real morphology of these complex particles, by using a 2D disc as a proxy.

o Lines 349-350: There exists some confusion, as you state “the mollusk shell has a low convexity (a hight value convexity)” and then “is convex in 3D (a low value of convexity)”. Please amend, as to my understanding, you try to say that the 2D projections of the shell demonstrate a high convexity (high values of convexity), although the real 3D shape is concave (has low values of convexity).

o 3rd Conclusion: Angularity has nothing to do with form. Please remove any reference to angularity unless if it is calculated (e.g. using the formulae of Kong and Fonseca, 2018). I think you meant to say “the grains were less equant (had a lower aspect ratio…)”.

o 3rd Conclusion: In 3D, the particles were more concave (less convex), since they exhibited lower convexity values. Please revisit this statement.

O 5th Conclusion: Please rephrase to “concave” (i.e. not convex). A 3D particle can be concave in 3D and its 2D projections can be convex. The inverse is not possible, i.e. for a convex 3D object to have non-convex 2D projections, as it is currently stated.

o 7th Conclusion: The scatter around all the linear regression lines seems just too much. I am not convinced that all these shape parameters are related or by the provided reason "This is possibly due to the biogenic nature of the soil", as evidence has not been presented to support such statement. From a theoretical standpoint, I agree that sphericity is expected to have come correlation to aspect ratio, as both represent some degree of "compactness" or "equancy" (i.e. non-flatness and non-elongation). Also, as I noted, the aspect ratio is equal to Aspect Ratio=Flatness*Elongation (using the Zingg formulae used in the manuscript), so two out of these 3 indices are independent. My advice would be to either provide reasons/a hypothesis of causality regarding why two indices would be correlated or else avoid a statement that is not well-substantiated. Also, I do not agree that 3D sphericity is correlated to convexity, as the value of the correlation factor (0.53) is not very compelling. Conceptually, why should more "equant" particles be more "convex"? Flat or elongated particles can be convex or concave as well, as demonstrated nicely in Figure 2.

A general comment:

o For particles with high intragranular voids, such as the analysed calcareous sand, using Feret dimensions might not be the best solution, as these can be non-representative of the actual particle form. Calculating the particle dimensions using a bounding box might be a good approach for materials with "bulky" particles, without intragranular voids. For instance, the minimal bounding box of an object has the same minimal bounding box with its convex hull, though these can be significantly different from a morphological standpoint. This limitation could be introduced in the manuscript. Have the authors considered shape indices that avoid using a simplified shape (i.e., a bounding box), e.g. based on spherical-harmonics (e.g. https://doi.org/10.1680/jgele.17.00011) or tensor-based indices (e.g. such as in http://doi.org/10.7148/2020-0256)?

Some minor/editorial comments:

o Line 30: Please correct the typo: “Advanced imaging techniques are …”

o Line 33: Please remove the extra bracket “(”.

o Lines 49-52: Please rephrase this sentence to make the message clearer.

o Line 54: “for calcareous…”.

o Line 62: “A selection…” I would start a new sentence.

o Line 63: “A summary …” would be more appropriate.

o Line 114: Please fix typo “3D”.

o Line 135: Please rephrase: “from the segmented the DIA imaging…”.

o Line 160: Please remove the double reference to “d_ESD”.

o Line 183: Please amend to “μCT”.

o Line 190: Fix broken sentence “sieve of the and the …”.

o Line 195: Update the broken hyperlink.

o Line 203: Replace typo "my" with "by".

o Table 3 is shown before Table 2, which is irregular.

o Table 3. Please add a header before the units, e.g. “Sieve size (mm)”.

o Line 250: Typo “range”.

o Line 351: Typo “bioclast”.

o Line 412: Please correct: AR_3D: Three-dimensional (3D) aspect ratio.

o Line 431: Please fix typo ”3D”.

Author Response

Please see attachment. Response to all reviewer comments is also included in the cover letter.

Reviewer 2 Report

In this article, the authors compared measurements of particle shape parameters from 2D DIA & 3D XRCT. It’s an interesting subject, however, it seems that there is no sufficient new conclusions compared to the litterature as follows:

  • Li et al., 2021 (Comparison of 2D and 3D dynamic image analysis for characterization of natural sands) confirmed the efficacy of Dynamic Image Analysis (DIA) for evaluating particle size and shape parameters and that 3D DIA is superior to 2D DIA.
  • Li et al., 2021 (Efficacy of 3D Dynamic Image Analysis for Characterizing the Morphology of Natural Sands) compared sand particle size and shape descriptors obtained using both 3D DIA and µCT for three natural sands having wide granulometries.

In my opinion, it makes more sense if this article was publised before Li et al., 2021 (Efficacy of 3D Dynamic Image Analysis for Characterizing the Morphology of Natural Sands).

Besides, some conclusions seem to be evident due to the difference between 2D & 3D measurements and some parameter correlations seem to be forced as the point clouds are dispersed.

Author Response

(The authors gave the same response as above.)

Reviewer 3 Report

This work studied and compared the results of 2D and 3D shape descriptors that are obtained from 2D DIA and 3D X-Ray CT techniques, respectively; and thereby demonstrated the need of 3D X-Ray CT technique for shape acquisition and characterization. The reason that resulted in the difference in 2D and 3D shape descriptors is furthered discussed. The main limitation, in the reviewer’s opinion, is the potential engineering application or implication of this work. Except for that, this work could be suggested for publication after resolving the following comments.

  1. This work aims at a broad sense of sand particles, including Bioclast, Bryozoan, Bioclast, and mollusk shell. It is suggested to highlight the motivation and potential engineering application or implication of the results of the shape analysis in this work.

  1. Eq. 9, it is suggested to adopt a definition of 3D sphericity that is comparable with the 2D case. For example, in 2D, the sphericity is the ratio of the perimeter of equal-area sphere to the perimeter of the particle. A similar definition is available in 3D, as that is used in Lai & Chen (2019). Please also refer to Angelidakis et al. (2021) for a detailed review on the shape descriptors.

Lai, Z., & Chen, Q. (2019). Reconstructing granular particles from X-ray computed tomography using the TWS machine learning tool and the level set method. Acta Geotechnica, 14(1), 1-18.

Angelidakis, V., Nadimi, S., & Utili, S. (2021). SHape Analyser for Particle Engineering (SHAPE): Seamless characterisation and simplification of particle morphology from imaging data. Computer Physics Communications, 265, 107983.

  1. line 195, reference not typeset properly.

  1. The Particle size curves in Figure 5 are confusing. Please consider removing or elaborating the description in the caption and main text.

  1. The shape analysis would be more insightful if it is based on different categories of sand particles (e.g., Bioclast, Bryozoan, Bioclast, and mollusk shell, as shown in Fig. 2), rather than mixed ones.

Author Response

(The authors gave the same response as above.)
